# Social Isolation Does Not Alter Exploratory Behaviour, Spatial Learning and Memory in Captive Damaraland Mole-Rats (*Fukomys damarensis*)

**DOI:** 10.3390/ani13030543

**Published:** 2023-02-03

**Authors:** Arantxa Silvia Blecher, Maria Kathleen Oosthuizen

**Affiliations:** 1Department of Zoology and Entomology, University of Pretoria, Private Bag X20, Pretoria 0028, South Africa; 2Mammal Research Institute, Department of Zoology and Entomology, University of Pretoria, Private Bag X20, Pretoria 0028, South Africa

**Keywords:** Bathyergidae, captive, colony, housing conditions, social isolation, learning, open-field test, Y-maze

## Abstract

**Simple Summary:**

Good exploration and navigational skills are important for subterranean mammals because they inhabit an energetically costly environment. Damaraland mole-rats are social animals, but in captivity, they can be housed in colonies or individually. Social isolation may have effects on their exploratory behaviour and memory performance. Overall, colony-housed and individually housed mole-rats explored readily, but single queens and non-breeding females were more active, possibly related to increased anxiety-like behaviour when housed individually. All groups, except single-housed non-breeding females, also showed improvements in solving a Y-maze, while both colony and single males made fewer errors in the maze over time. Thus, learning is probable but may have been underestimated in this experiment. In general, social isolation has a limited effect on Damaraland mole-rat exploration and spatial learning.

**Abstract:**

Exploratory behaviour, spatial learning and memory affect the survival of animals and appear to be dependent on the specific habitat that a species occupies. Good spatial navigation and memory are particularly important for subterranean animals, as it is energetically expensive to inhabit this niche. Damaraland mole-rats are subterranean mammals that live in colonies with organised social structures. Damaraland mole-rats have been maintained in the laboratory for many years and can be housed in groups or individually. We evaluated the effect of social isolation on the exploratory behaviour and spatial memory of single-housed and colony-housed animals and also considered potential differences in animals with different social statuses. We predicted that solitary housing would increase anxiety-like behaviour and result in higher activity and more errors when solving a maze. Exploration by colony- and single-housed mole-rats was tested in an open-field test, where all individuals explored readily. Single-housed queens and non-breeding females showed increased activity and spent more time in tunnels, which can be explained by increased anxiety. In the Y-maze, improvements in solving the maze were observed in all experimental groups, except in single-housed non-breeding females. In addition, all males showed a decrease in the number of errors in the maze. Spatial learning is thus apparent but could not be conclusively proven. It was possibly underestimated, as magnetic cues that may be used by mole-rats as stimuli for navigation were removed in the experimental setup. Overall, it appears that social isolation has a limited effect on the exploratory behaviour and spatial learning of Damaraland mole-rats.

## 1. Introduction

Exploratory behaviour includes reactions, as well as triggered behaviours, of animals to new experiences or environments [1,2]. It allows individuals to gain information and become familiar with their environment [1,3]. By exploring their environment, animals locate food and water resources, as well as hiding places [3,4]. 

Exploratory behaviour can be influenced by various factors from an animal’s internal and external environment. This includes environmental factors such as resource availability and knowledge of alternative resources, individual factors such as cognitive capacity and physical strength, and social factors such as competition [5]. Exploration may also be dependent on environmental complexity, light conditions, genetic background, age, sex, housing conditions and social and reproductive states [1,6,7,8]. Often, exploration is also linked to other behaviours, including aggression [9], dispersal [2,10], predator avoidance [11] and foraging [12].

In addition to exploration, survival is also dependent on navigation, spatial learning and memory [13]. Efficient navigation aids in finding food or mates and guides migration over long distances [14,15,16]. This is particularly important for subterranean animals, such as mole-rats, as they inhabit a harsh environment with considerable costs associated with living there [17]. For example, digging through soil is energetically expensive; thus, new burrow formation, travelling and dispersal become costly [13,17]. Good navigation and spatial learning allow subterranean animals to reduce energy expenditure and avoid predators, while successfully foraging and finding mates [18]. Costanzo, Bennett and Lutermann [13] found generally high memory retention in two subterranean mole-rat species, suggesting a strong selective pressure on the spatial ability enforced by their lifestyle.

Spatial navigation and memory may be influenced by several factors. It has been suggested that the Earth’s magnetic field may affect navigation and facilitate staying on course during digging [19]. In addition, the social structure of a species (solitary or group living) has been shown to affect the learning abilities of animals. For example, a social mole-rat species has been shown to have superior spatial learning and memory performance compared to a solitary species [13]. Environmental conditions, such as being in captivity or free-living, can also have differential effects on memory, but this seems to be dependent on the species and its social structure. Social wild Natal mole-rats (*Cryptomys hottentotus natalensis*) outperformed captive animals in spatial navigation tests [15], while there were no particular differences in spatial navigation ability between wild and captive solitary Cape mole-rats (*Georychus capensis*) [20]. Stress was shown to disrupt spatial memory, as observed in rodents [21]. Lastly, the effect of sex on memory and spatial navigation ability is unclear and widely contested, with several hypotheses attempting to explain the evolution of sex differences [22]. 

Housing conditions in captivity can affect the exploration, spatial learning and memory of experimental animals. These conditions may differ in the type and size of housing, lighting, ambient temperature, air quality and enrichment. Moreover, the housing of laboratory animals should consider the general biology and life history of the animals under investigation, for example, whether they are solitary or social. Social animals should be housed in groups, unless the experimental design dictates otherwise, since the isolation of social animals can cause stress [23,24,25]. 

Housing conditions were previously shown to have considerable effects on exploratory behaviour [26,27], general activity [8,28,29], stress and anxiety [30,31], as well as energy balance and metabolic health [29]. Furthermore, social isolation affects memory, anxiety and neurogenesis in the brains of laboratory rats [32]. Hannibal et al. [33] showed that in macaques, acclimation to social separation may take months and can impact research results. Therefore, it is important to take housing conditions, especially those of social species, into account when investigating aspects of behaviour, learning and memory. 

African mole-rats from the family Bathyergidae are contained in six genera: three are solitary-dwelling, and three are social. Two of the social genera contain species that are highly social (eusocial), with the Damaraland mole-rat (*Fukomys damarensis*) often classified as eusocial [17,34]. All mole-rats are subterranean, which means that they require good navigation, learning ability and memory to successfully inhabit this costly environment [17,18]. 

Damaraland mole-rats live in highly organised social groups, and they are cooperative breeders, with their reproduction distributed unequally among members of the group [35,36]. A group usually consists of a single reproductive female (the queen), that reproduces with the most dominant males, and non-reproductive males and females [36,37]. Non-breeding females are anovulatory and physiologically suppressed by the presence of the queen, while non-reproductive males are physiologically similar but behaviourally different to reproductive males [36,37]. The non-reproductive individuals act as helpers, participating in behaviours such as burrow maintenance, tunnel sweeping and the carrying of food and nest material [34,35]. 

There are limited studies on the physiological and behavioural changes that occur in Damaraland mole-rats when they are removed from their natal colony and housed individually. In these existing studies, it is unclear when these changes occur or whether they remain constant over extended periods of removal. Some studies have shown hormonal changes, where the progesterone levels of females increased once they were removed from the colony, indicating that they became reproductively active [38,39,40]. In terms of Damaraland mole-rat behaviour, research has thus far only involved comparisons between animals within the same colony with different social statuses [10]. To date, no studies have investigated exploration and memory in single-housed Damaraland mole-rats. 

The objective of our study was to determine whether the housing conditions (colony- or single-housed) of captive social Damaraland mole-rats would affect their exploratory behaviour and reference memory. We anticipated that animals that are housed individually exhibit more anxiety-like behaviour by exploring more but perform worse in learning tasks. We were also interested in whether there are differences in exploratory behaviour and reference memory between the reproductive female (queen) and the non-reproductive males and females within the groups.

## 2. Material and Methods

### 2.1. Study Animals

The study animals are part of laboratory colonies at the University of Pretoria, Pretoria, South Africa, and have been in captivity for several years. For the current study, two groups of animals were used. One group of mole-rats consisted of animals housed in colonies, while animals in the other group had been housed individually for at least two years. The latter was previously used for other non-invasive behavioural experiments where single housing was required and thus have been isolated for more than two years, also allowing easy comparison with solitary species. All mole-rats that were individually housed were adults at the time of removal from their natal colonies, and all animals were separated on the same day. Individuals in the two groups were categorised depending on their breeding status, namely, breeding females (queens) and subordinate animals (both non-breeding males and non-breeding females). The categories of animals that were housed individually were assigned based on the breeding status they had while they were still in a colony. Queens were identified by the presence of nipples and a perforate vagina. 

The colony group consisted of 29 animals (7 queens, 8 subordinate females, 14 subordinate males), and the individually housed group consisted of 17 animals (6 queens, 4 subordinate females, 7 subordinate males). The body weights of experimental animals were monitored and remained stable for the duration of the study. 

The mole-rats were housed in plastic crates with the bottoms layered with wood shavings. Additionally, they received a plastic pipe and tissue paper for nesting material. They were fed sweet potato that was occasionally supplemented with carrots and apples. No water was supplied, as enough water is derived from their food. Animals were housed at a constant temperature of 25 °C and with a fluorescent overhead light at 150 lux at ground level. Experimental procedures took place between 08h00 and 15h00 during the day. All mole-rat species have somewhat variable daily rhythms, with both nocturnal and diurnal individuals and some animals with no apparent rhythms in the same species [41,42]. All animals were awake before experimentation, and none of the experimental animals appeared sleepy or lethargic during the experiments. Therefore, we are confident that the time of day did not have significant effects on the study. All experimental procedures were approved by the Animal Use and Care Committee at the University of Pretoria (EC013-09). 

### 2.2. Experiment 1: Exploratory Behaviour in a Tunnel Maze

Exploratory behaviour was tested in a modified open-field box. This white wooden box had an open top and dimensions of 20 × 40 × 20 cm. On one of the long sides of the box, three black plastic tunnels (20 cm long and 7 cm in diameter) were attached. These tunnels were spaced at equal distances from one another and positioned 1 cm above the floor of the box. Additionally, the floor of the box was covered with a thin layer of wood shavings. Between tests on different animals, wood shavings were changed, and the box and tunnels were cleaned with 95% alcohol and distilled water [43]. 

Animals were placed in the centre of the box, and their behaviour was recorded for three minutes. Specifically, four different measures were recorded: the latency to enter the first tunnel, the total duration spent in tunnels, the total number of entries into tunnels (a count of how many times an animal enters any tunnel) and the number of tunnels (out of the three available tunnels) visited by the mole-rats. A tunnel entry was only counted if all four feet of the mole-rat were inside a tunnel [43]. Animals that showed a longer latency to enter the first tunnel, spent a shorter time in tunnels and made more tunnel entries were considered more exploratory. 

### 2.3. Experiment 2: Spatial Memory in a Water Y-Maze

For this experiment, a Perspex Y-shaped maze was used. Each arm of the maze measured 50 × 10 × 20 cm and ended in an escape hole, 2 cm above the floor, with an attached tunnel. The ends of each arm could be closed off with a guillotine door [44]. 

The bottom of the Y-maze was covered with about 1 cm of room-temperature water, used to motivate the animals to move throughout the maze. For each test, one arm of the maze was chosen as the correct arm, which was opened at the end into a Perspex tunnel connected to a dry nest box. The other two arms (wrong arms) ended in blocked Perspex tunnels so that these arms would look identical to the correct arm from the centre of the maze. Of these wrong arms, one was chosen as the start arm into which the animals were released, facing the blocked tunnel. 

The arms of the maze were labelled A, B and C. Arm A was the start arm for the first trial, while arm B was the wrong arm, and arm C was the correct arm. After that, the start arm was chosen randomly between arms A and B so that half of the trials per day started in each arm. Furthermore, the correct arm with the escape hole to the nest box was always located to the right of the start arm. For example, when the starting position was in arm A, the nest box was connected to arm C, and if the starting position was in arm B, the nest box was connected to arm A. 

After the mole-rats were released, they were allowed a maximum of one minute to find the escape hole and complete the maze. Both the time to complete the maze and the number of wrong turns taken by the mole-rat were recorded [44]. If an animal did not complete the maze in under one minute, it was awarded a time of 60 s. Again, escapes were only recorded when all four feet of the animal were inside the tunnel leading to the nest box. Each mole-rat underwent ten trials per day for four consecutive days. Additionally, the time between each consecutive trial varied between 45 and 60 min. 

The mass of the animals was observed to remain stable during all experimental procedures (data not shown). All experiments were recorded with an overhead video recorder and analysed manually after the completion of the experiments. 

### 2.4. Statistical Analysis

All statistical analyses were performed using R statistical software [45]. Significance was set at *p* < 0.05, and all results are shown as mean ± standard error. For both experiments, data were not normally distributed (tested with the Shapiro–Wilk normality test). For Experiment 1, parameters for exploratory behaviour were analysed using generalised linear models from the R package lme4 [46]. A gamma distribution with a log link function was used in the model for latency to enter the first tunnel and the total duration in tunnels, while a Poisson distribution with a log link function was used for the analysis of the total number of entries into the tunnels and the number of tunnels visited. Furthermore, 1 was added to all values of total duration to remove zeros from the data. Predictor variables were environment (colony-housed or single-housed), group (male non-breeder, female non-breeder or female queen) and their interaction. For Experiment 2, the mean time to complete the maze per day was calculated for each animal, while the number of errors per day was calculated by dividing the total number of errors for the ten trials by ten. The time to complete the maze and the number of wrong turns were analysed using generalised linear mixed models from the lme4 package [46] with a gamma distribution and a log link function. Environment, group, day and their interactions were used as fixed factors, and animal ID was used as a random factor to account for repeated measures. Furthermore, 1 was added to all values of the mean number of wrong turns to remove zeros from the data. The *p*-values for each model were generated using the package car [47]. Finally, Tukey’s HSD post hoc multiple comparisons were generated using the R package emmeans [48] for all post hoc tests in both experiments.

## 3. Results

### 3.1. Experiment 1

The latency to enter the first tunnel was significantly influenced by the environment (χ^2^ = 5.985, df = 1, *p* = 0.014) and the interaction effect between the environment and group (χ^2^ = 6.010, df = 2, *p* = 0.050). Nevertheless, none of the interaction pairwise comparisons were significant (Figure 1 and Appendix A). The total duration spent in tunnels was significantly influenced by the interaction effect between the environment and group (χ^2^ = 7.038, df = 2, *p* = 0.030); however, post hoc tests revealed no significant pairwise comparisons (Figure 1 and Appendix A). The total number of tunnel entries was significantly affected by the environment (χ^2^ = 4.360, df = 1, *p* = 0.037) and the interaction effect between the environment and group (χ^2^ = 12.8459, df = 2, *p* = 0.002). Single queens had a significantly higher number of total tunnel entries than colony non-breeding females, colony queens, colony males and single males (Table 1; Figure 2 and Appendix A). Lastly, the number of tunnels visited by the animals was not significantly influenced by the environment, group or interaction term (Figure 2 and Appendix A).

### 3.2. Experiment 2

The time to exit the maze was significantly influenced by the day (χ^2^ = 253.412, df = 3, *p* < 0.001) and the three-way interaction between the environment, group and day (χ^2^ = 18.545, df = 6, *p* = 0.005). Post hoc comparisons revealed 14 meaningful comparisons (Table 2, Figure 3). The number of wrong turns was significantly influenced by the day (χ^2^ = 91.269, df = 3, *p* < 0.001), the two-way interaction term between the group and day (χ^2^ = 16.106, df = 6, *p* = 0.013) and the three-way interaction between the environment, group and day (χ^2^ = 12.907, df = 6, *p* = 0.031). Post hoc analysis showed six meaningful comparisons (Table 3; Figure 4).

## 4. Discussion

We assessed the exploratory behaviour and spatial learning of breeding and non-breeding Damaraland mole-rats that were housed in colonies and singly. We expected that solitary housing would increase the anxiety-like behaviours of animals, thereby increasing activity, while we anticipated that this group would show increased wrong turns in the Y-maze during the learning experiment. 

### 4.1. Exploratory Behaviour

The time variables during exploration were similar in mole-rats with different social statuses and housing conditions. This suggests that both single- and colony-housed queens and non-breeding mole-rats explored readily and had no preference between the open area and the tunnels. Similarly, wild and captive solitary Cape mole-rats also readily explore open spaces and tunnels, with no difference between the two groups [20]. However, wild Damaraland mole-rats showed variation among groups of different social statuses within the colonies. Worker animals entered tunnels faster than queens, and the mean time spent in tunnels was higher for workers, while the total duration in tunnels was similar between workers and queens [10]. 

The number of tunnels visited was similar in all animals, while single-housed queens entered the tunnels more frequently compared to all other groups, except single non-breeding females. Therefore, it appears that single-housed queens, and to a lesser extent single-housed non-breeding females, are more active in the modified open-field arena. When compared to Cape mole-rats with a similar experimental setup, Damaraland mole-rats in the present study showed more tunnel entries and a higher number of tunnel visits [20]. Oosthuizen [10] found no differences in the number of tunnel entries or number of tunnels visited between wild-captured Damaraland mole-rat workers and queens.

The higher activity seen in single-housed females may be attributed to anxiety. The social isolation of social species has been shown to increase stress in the closely related naked mole-rat [24], as well as in other rodents [23,49] and primates [25,50]. Additionally, social isolation increases activity, as observed in mice and rats [8,51,52,53]. Therefore, the isolation of single queens and non-breeding females may have increased their stress levels, and thus, they became more active and entered tunnels more frequently. These results are in agreement with the notion that the colony housing of social species provides a level of enrichment and comfort for the animals, emphasising that social species should not be housed individually. Single-housed Damaraland mole-rat males, on the other hand, appear to be less anxious and thus less active. This is similar to what has been observed in naked mole-rat males that were removed from their colonies [24]. It is possible that males are less invested in colony life and are more likely and faster to disperse when conditions are suitable. 

With the exception of the total number of tunnel entries, all measured parameters of exploratory behaviour were similar in colony-housed and single-housed animals. The social isolation of Damaraland mole-rats thus does not appear to affect their exploratory behaviour. This is in contrast to other rodent species, such as mice and rats. Previous studies showed increased activity in single-housed mice [27,51,52] and rats [26]. The effect of housing conditions on exploratory behaviour may, therefore, be species-specific. The period of isolation could also play a role, as Garzón and Del Río [51] observed marked behavioural changes in mice after 10 months of isolation. In our study, animals in the single-housed group were isolated for about two years before testing; therefore, we could speculate that social isolation is not overly stressful for Damaraland mole-rats in the laboratory, where they have access to ad libitum food and have a relatively quiet environment to live in. The sociality and large colony sizes of Damaraland mole-rats specifically are thought to be driven by reduced dispersal opportunities as a result of their arid habitat [54]. 

### 4.2. Spatial Memory

Most of the experimental groups learned quickly and showed a significant improvement in finding the escape hole by the second day compared to the first day. Exceptions were the colony-housed queens, which only showed a significant improvement on the third day, and the single-housed non-breeding females. The latter did not improve their time to find the escape hole at all; however, they started off with a shorter completion time compared to the other groups, and the time of completion was comparable to that of the other groups on the second, third and fourth days. They did, however, show a slight increase in time over the experimental period. Wild-trapped Damaraland mole-rats housed in their colonies also showed a fast improvement in time (one day) in all groups apart from disperser females (two days) [10]. The social isolation of a social animal can be stressful, especially for females [24,31]. Moreover, stress-related physiological responses to social isolation reduce hippocampal neurogenesis, important for cognitive ability, and this effect seems to be greater in females compared to males [32,55]. Therefore, the single-housed non-reproductive females in the present study could have manifested a stress-related reduction in cognitive ability, while the effect of stress was not big enough in the single-housed males to have an effect. Although the colony queens in our study were slower to become familiarised with the maze, the overall profile of their improvement is similar to the other groups. They show a similar trend to wild disperser females, which also appear to learn more slowly [10]. The slower learning curve in our current females may be a result of a slightly faster time on the first day. However, their times to complete the maze on any of the four experimental days did not differ from those of any of the other groups and thus are not meaningful in the context of this study. 

Overall, both captive and wild Damaraland mole-rats appear to learn more quickly than solitary Cape mole-rats [10,20]. Damaraland mole-rats significantly improve their time to exit the maze within one day, whereas the solitary species only shows an improvement over two days. The superior learning and memory performance of Damaraland mole-rats compared to Cape mole-rats was also observed by Costanzo, Bennett and Lutermann [13]. The difference between the two species could be explained by their disposition, with Cape mole-rats being more aggressive and possibly more anxious, which in turn affects their cognitive function [20,30]. Alternatively, differences in habitat and relative tunnel lengths could necessitate the superior learning ability of Damaraland mole-rats compared to Cape mole-rats, where the burrow systems are more complex in the social species [13,54,56]. 

The number of wrong turns taken in the Y-maze did not yield a particular pattern for group- or single-housed queens and non-breeding females. The latter specifically showed erratic behaviour, with performance increasing and then decreasing again. In contrast, colony males significantly improved from day one to day two, making fewer wrong turns and then remaining stable through days three and four. Single males also showed an improvement, but one day later on day three, and then remained stable on day four. Oosthuizen [10] previously observed that wild-trapped colony Damaraland mole-rats made the fewest wrong turns on the first day in the same Y-maze experiment compared to the next three days. Furthermore, the study also showed that female dispersers made more wrong turns than male dispersers, while worker males and females made equal numbers of wrong turns [10]. In contrast, captive solitary Cape mole-rats showed no decrease in the number of errors on consecutive days [20]. 

Thus, males appear to be better at solving the Y-maze, making fewer wrong turns than females and improving the time to escape the maze. This may again be anxiety-related, with females overall exhibiting higher stress levels than males, which could impair their performance [32]. However, this phenomenon was also observed in one of the female groups of wild Damaraland mole-rats [10]. It was suggested that females employ different dispersal strategies than males and, in this context, were less motivated to complete the learning task [10]. While some studies report sex differences in learning and memory performance in rodent species [18,57], others do not [58,59]. Notably, Costanzo, Bennett and Lutermann [13] found no sex differences in the performance of Damaraland mole-rats; however, their experimental setup differed from ours in that they provided a food incentive after starving animals for 24 h. By providing a food incentive, we could potentially have eliminated sex differences in our results. 

In general, the number of wrong turns made by the mole-rats may be used as a proxy for spatial learning, with fewer errors on consecutive days indicating that animals learned where the escape hole was located. In the present study, only males made fewer errors on consecutive days, suggesting that some learning was evident, but our results do not conclusively indicate that spatial learning was achieved. During the experiment, the orientation of the Y-maze was changed between trials, thereby changing the global cues present outside of the maze. This prevented the animals from using outside spatial information, for example, geomagnetism, as a cue for navigation. Mole-rats have poor vision [60], and it has been suggested that they possess the ability to detect magnetic stimuli for spatial orientation in their subterranean environment [19]. As suggested by Oosthuizen [10], the removal of magnetic cues as potential stimuli may have made spatial learning more difficult for the mole-rats in the present study, and their spatial learning ability could have been underestimated.

Overall, the average number of errors for each day was low (>1) in all groups and comparable to those observed previously in this species [10] and captive Cape mole-rats [20]. In contrast, wild-trapped Cape mole-rats displayed a higher number of errors in the same experimental setup [20]. This difference may again be attributed to the difference in their burrow systems. 

## 5. Conclusions

In conclusion, all Damaraland mole-rats in our study explored readily when exposed to a novel environment, and exploration appears to be similar for colony-housed and single-housed individuals. Single-housed queens and non-breeding females displayed higher activity, which could be associated with a higher level of anxiety. Empirical studies measuring glucocorticoid levels would be required to categorically prove this. The time to find the escape hole improved in all experimental groups except in single-housed non-breeding females, where no improvement was evident, but they were faster than the other groups on the first day. In addition, the number of errors in the maze did not decrease, except for single-, and colony-housed males. Therefore, spatial learning is apparent but could not be conclusively proven in captive Damaraland mole-rats under the current experimental conditions. The social isolation of captive Damaraland mole-rats appears to have a muted effect compared to other social laboratory rodents, as it did not affect their exploratory behaviour or spatial learning abilities. This may be related to their social evolution and the factors driving sociality in the Damaraland mole-rat.

## Figures and Tables

**Figure 1 animals-13-00543-f001:**
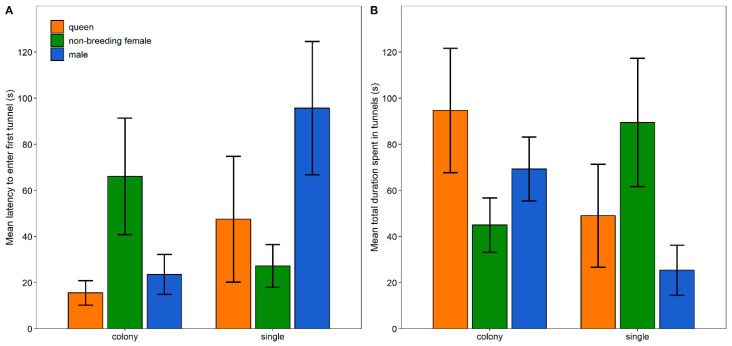
Latency to enter the first tunnel (±SE) (**A**) and total duration spent in tunnels (±SE) (**B**) by colony- and single-housed Damaraland mole-rats in an open-field box.

**Figure 2 animals-13-00543-f002:**
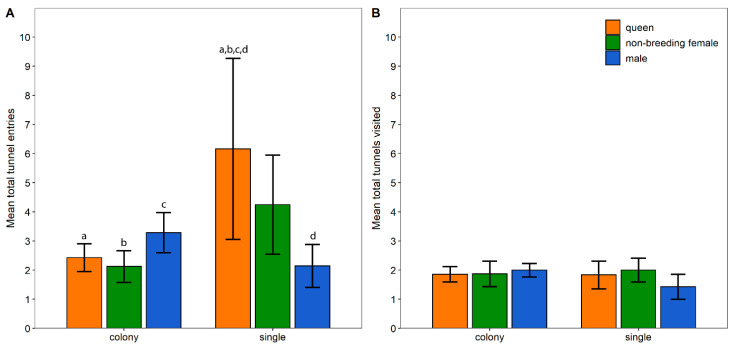
Total tunnel entries (±SE) (**A**) and total number of tunnels visited (±SE) (**B**) by colony- and single-housed Damaraland mole-rats in an open-field box. Significant differences between pairs are indicated by letters above the bars (*p* < 0.05).

**Figure 3 animals-13-00543-f003:**
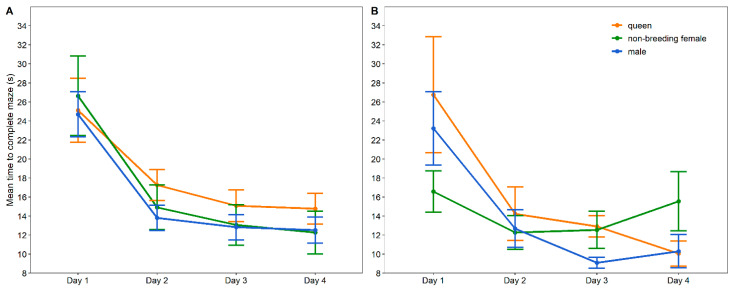
Mean time (±SE) taken by colony- (**A**) and single-housed (**B**) Damaraland mole-rats to complete a Y-shaped maze on four consecutive days. Significant differences between the different groups are indicated in Table 2.

**Figure 4 animals-13-00543-f004:**
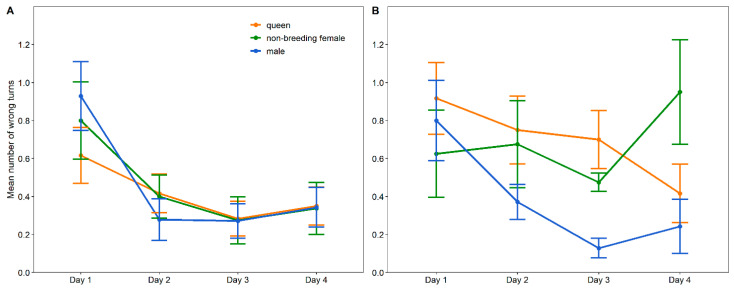
Mean number of wrong turns (±SE) made by colony- (**A**) and single-housed (**B**) Damaraland mole-rats in a Y-shaped maze on four consecutive days. Significant differences between the different groups are indicated in Table 3.

**Table 1 animals-13-00543-t001:** Post hoc significant differences in the mean total number of tunnel entries between experimental groups in Experiment 1.

Contrast	Mean ± SE	Contrast	Mean ± SE	*p*-Value
Single queens	6.17 ± 3.11	Colony non-breeding females	2.12 ± 0.55	0.004
		Colony queens	2.43 ± 0.48	0.018
		Colony males	3.29 ± 0.69	0.050
		Single males	2.14 ± 0.74	0.007

**Table 2 animals-13-00543-t002:** Significant post hoc differences in the mean time to complete a Y-shaped maze in Experiment 2 between experimental day 1 and days 2, 3 and 4.

Contrast	Mean ± SE	Contrast	Mean ± SE	*p*-Value
Colony non-breeding females day 1	26.64 ± 4.18	Colony females day 2	14.91 ± 2.35	<0.001
		Colony females day 3	13.06 ± 2.14	<0.001
		Colony females day 4	12.26 ± 2.26	<0.001
Colony queens day 1	25.12 ± 3.38	Colony queens day 3	15.08 ±1.67	0.032
		Colony queens day 4	14.77 ± 1.63	0.016
Colony males day 1	24.69 ± 2.37	Colony males day 2	13.80 ± 1.33	<0.001
		Colony males day 3	12.83 ± 1.33	<0.001
		Colony males day 4	12.52 ± 1.37	<0.001
Single queens day 1	26.75 ± 6.10	Single queens day 2	14.24 ± 2.82	0.001
		Single queens day 3	12.92 ± 1.11	<0.001
		Single queens day 4	10.05 ± 1.32	<0.001
Single males day 1	23.21 ± 3.86	Single males day 2	12.69 ± 1.97	<0.001
		Single males day 3	9.09 ± 0.56	<0.001
		Single males day 4	10.30 ± 1.76	<0.001

**Table 3 animals-13-00543-t003:** Significant post hoc differences in the mean number of wrong turns in Experiment 2 between experimental day 1 and days 2, 3 and 4.

Contrast	Mean ± SE	Contrast	Mean ± SE	*p*-Value
Colony non-breeding females day 1	0.80 ± 0.20	Colony non-breeding females day 3	0.28 ± 0.12	0.020
Colony males day 1	0.93 ± 0.18	Colony males day 2	0.28 ± 0.12	<0.001
		Colony males day 3	0.27 ± 0.09	<0.001
		Colony males day 4	0.34 ± 0.10	<0.001
Single males day 1	0.80 ± 0.21	Single males day 3	0.13 ± 0.05	<0.001
		Single males day 4	0.24 ± 0.14	0.015

## Data Availability

Data is available on request.

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
