# Peer review of "Social Isolation Does Not Alter Exploratory Behaviour, Spatial Learning and Memory in Captive Damaraland Mole-Rats (Fukomys damarensis)"

_animals, 2023, doi:10.3390/ani13030543_

Round 1

Reviewer 1 Report

Blecher and Oosthuizen present the results of a very interesting and relevant study on the effects of social isolation on exploratory behavior and cognitive performance in Damaraland mole rats of different social statuses and sex. Exploration was evaluated in a modified open-field test and cognitive performance was tested in a Y-maze test. Overall, their results indicate minor effects of social isolation on exploration and spatial memory, depending on sex or social status. 

The study is well and comprehensively written, however, I do have some comments and suggestions the authors might want to consider.

With regard to the modified open-field test, I'm unsure if the study design and data analysis allow getting a conclusive picture of exploratory behavior in the mole rats. Specifically, using the number of tunnel entries and time spent in the tunnels as proxies of exploration may not be sufficient to fully describe exploratory behavior given the differences in the latency to enter a tunnel without providing further information on the animal's behavior in the open space (resting vs locomotion).

For instance, single males have a long latency to enter a tunnel but have a lower number of tunnel entries compared to group-housed individuals. This is somewhat contradictory given that the authors suggest that both longer latency and a higher number of tunnel entries indicate a higher level of exploration.

Usually, additional parameters like total distance traveled are used to assess exploration. Thus, adding information on locomotor activity (moving vs resting) when the animals are in the open space of the arena would give valuable insights into their exploratory behavior.

Also, the difference between the total and mean duration in the tunnel is not clear to me, the authors may specify why they made the separation and how it was analyzed. I believe that the proportion of time spent in the tunnel would be a better estimate and I suggest the authors add this parameter.

Minor:

page1 line 5: remove "and" from the author list

page 1 line 14 (and following): I suggest replacing "anxiety" with "anxiety-like behavior" 

page 3 lines 127 - 133: Please provide more insights about the age of the experimental animals and the time when individually-housed animals were separated from the colony (age at separation). Discuss possible age effects of social isolation. What is the min and max time of social isolation in your group and does that affect the results?

page 3 line 138: add information on the light and temperature conditions the animals have been kept in.

page 4 line 155: please specify if mean total tunnel visits refer to the visit of different tunnels or if that also includes visiting the same tunnel repeatedly. It could be added as additional information on how many of the 3 tunnels very explored by the animals.

page 4 line 155 (and following): information on the time when these experiments have been conducted is missing for both experiments. Exploration, anxiety-like behavior, and cognitive performance vary throughout the 24-h day. Please explain and discuss the potential effects of daytime on these experiments.

page 6 Figs 1 + 2: Please add individual data points to your bar graphs.

pages 7 + 8: Figs 3 + 4: labels X-axis missing

page 9 lines 293 - 295: please underpin statement of higher activity with data on locomotion in the open space (see comment above)

page 10, line 368: add a point at the end of the sentence

page 10, line 380: please re-phrase. Further evaluation is required to conclude the animals display higher levels of anxiety-like behavior. The results of this study are inconclusive.

Reviewer 2 Report

Overview

The goal of this study is to assess the effect of social isolation on the exploratory behavior and spatial memory in Damaraland mole-rats maintained in the laboratory for many years. For this goal, the authors examined exploratory behaviors in a tunnel maze (Experiment 1) and spatial memory in a water Y-maze (Experiment 2), and compared data obtained from single-housed and colony-housed animals of three social statuses, i.e., queen females, non-reproductive females, and non-reproductive males. The authors’ overall conclusion is that social isolation in this study had limited effects on the exploratory behavior and spatial memory in Damaraland mole-rats, although activity of single-housed females increased in Experiment 1 and spatial learning was apparent (but could not be conclusively proven) in Experiment 2.

All data in this study are rare and valuable to biological sciences. I have the following some comments for improvement.

01_page 3, lines 127─128: I think that it is necessary to explain more precisely why the authors used animals housed singly for more than two years, because it seems to me that social isolation for days or weeks can induce more drastic effects of solitary housing.

02_Material and methods: Please describe whether data analyses were conducted blind to experimental conditions or not. In addition, please describe whether data analyses were conducted automatically or manually.

03_ Material and methods: Please provide what time of day the authors conducted Experiments 1 and 2 (e.g., “between 09:00 and 17:00”).

04_Statistical analyses: Please provide more precise information about what statistical test was performed before the post-hoc statistical analysis. Did you perform statistical correction (e.g., Bonferroni correction) for multiple comparisons to avoid type 1 error? For example, the same statistical performance was repeated five times from one animal dataset in Experiment 1. Moreover, it appears that the authors misunderstood an interaction in statistics. I recommend asking advice of an expert in statistics.

05_page 8, lines 262─268: I think that it would be better to delete the following sentences. “Exploration is an important component of the behavioural repertoire of animal since >>>>>>>>>> vision cannot be used for navigation would also be beneficial.”

06_I think that there is a discrepancy between the following descriptions in the Introduction section and Discussion section.

The Introduction section (page 3, lines 118─119) “We anticipated that animals that were housed individually will be less exploratory and >>>>>>>>>>”

The Discussion section (page 8, lines 270─272) “We expected that solitary housing will increase the anxiety of animals, thereby increasing activity and time spent in tunnels in the exploratory experiment, >>>>>>>>>”

07_ page 9, lines 293─294: “The higher activity and exploration seen in the single-housed queens may be attributed to anxiety.” I think that it would be necessary to explain a correlation between high activity and high anxiety levels in this animal more precisely by citing references, because immobility such as freezing behavior is generally considered as a typical index of anxiety in laboratory rodents such as mice and rats.

08_ page 9, line 321: “(improved only on day three)” Is it true? How about day 4?

09_ page 9, line 322: “They did not improve their time to find the escape hole at all, however, >>>>>>>>>>” I cannot understand what “They” in the beginning of this sentence indicates, however, probably means “single-housed non-breeding females”. Please rephrase it more clearly.

10_ page 9, lines 336─337: “They did not show the same erratic behaviour compared to wild disperser females, thus our results may be coincidental.” It is impossible for me to understand what this sentence means. Please rephrase it more clearly, at least it should be necessary to cite some references.

11_ page 10, lines 349─358:

I consider that the sentence “None of the other experimental groups showed any changes in the number of wrong turns throughout the experiment. (lines 352─353)” is not true, because single-housed males showed fewer wrong turns on experimental days 3 and 4 than results on experimental day 1.

“Non-reproductive colony females and single males showed erratic behaviour, with performance increasing and then decreasing again. (lines 353─354)” Is it true?

Regarding this paragraph, I think that the authors can discuss sex differences in the number of wrong turns in the Y-maze test in Experiment 2, because both single-housed and colony-housed male Damaraland mole-rats, not females, showed decrease in the number of wrong turns on experimental day 4 in comparison to day 1.

12_Discussion: At the last part of the Discussion section, it would be better to comment on the study limitations of experimental design including potential sources of bias.

13_I recommend the following order of Figures and Tables: Figure 1 ─> Figure 2 ─> Table 1 ─> Figure 3 ─> Table 2 ─>Figure 4 ─> Table 3.

14_I question the need of Figure 1C. If possible, it would be better to delete it and related parts of your manuscript.

15_It seems to me that it would be better to rephrase table legends, for example,

“Table 1. Post hoc significant differences in the mean total number of tunnel entries between experimental groups in Experiment 1.”

“Table 2. Post hoc significant differences in the mean time to complete a Y-shaped maze in Experiment 2 between experimental day 1 and days 2, 3, and 4.”

“Table 3. Post hoc significant differences in the mean number of wrong turns in Experiment 2 between experimental day 1 and days 2, 3, and 4.”

16_Axis labels in Figures 3 and 4: I recommend changing horizontal axis labels “1, 2, 3, and 4” to “day 1, day 2, day 3, and day 4”. I also recommend deleting vertical axis labels in Figures 3B and 4B.

Round 2

Reviewer 1 Report

The authors of the study submitted a revised and improved version of the manuscript, with the appropriate feedback and changes according to the comments/ suggestions made on the earlier version of the manuscript. I recommend accepting the MS for publication. Congratulations to the authors!